# The problem with DDPG: understanding failures in deterministic environments with sparse rewards

## Abstract

In environments with continuous state and action spaces, state-of-the-art actor-critic reinforcement learning algorithms can solve very complex problems, yet can also fail in environments that seem trivial, but the reason for such failures is still poorly understood. In this paper, we contribute a formal explanation of these failures in the particular case of sparse reward and deterministic environments. First, using a very elementary control problem, we illustrate that the learning process can get stuck into a fixed point corresponding to a poor solution. Then, generalizing from the studied example, we provide a detailed analysis of the underlying mechanisms which results in a new understanding of one of the convergence regimes of these algorithms. The resulting perspective casts a new light on already existing solutions to the issues we have highlighted, and suggests other potential approaches.

## 1 Introduction

The Deep Deterministic Policy Gradient (DDPG) algorithm (Lillicrap et al. (2015)) is one of the earliest deep Reinforcement Learning (RL) algorithms designed to operate on potentially large continuous state and action spaces with a deterministic policy, and it is still one of the most widely used. However, it is often reported that DDPG suffers from instability in the form of sensitivity to hyper-parameters and propensity to converge to very poor solutions or even diverge. Various algorithms have improved stability by addressing well identified issues, such as the over-estimation bias in TD3 (Fujimoto et al., 2018b) but, because a fundamental understanding of the phenomena underlying these instabilities is still missing, it is unclear whether these ad hoc remedies truly address the source of the problem. Thus, better understanding why these algorithms can fail even in very simple environments is a pressing question.

To investigate this question, we introduce in Section 4 a very simple one-dimensional environment with a sparse reward function where DDPG sometimes fails. Analyzing this example allows us to provide a detailed account of these failures. We then reveal the existence of a cycle of mechanisms operating in the sparse reward and deterministic case, leading to the quick convergence to a poor policy. In particular, we show that, when the reward is not discovered early enough, these mechanisms can lead to a *deadlock* situation where neither the actor nor the critic can evolve anymore. Critically, this deadlock persists even when the agent is subsequently trained with rewarded samples.

The study of these mechanisms is backed-up with formal proofs in a simplified context where the effects of function approximation is ignored. Nevertheless, the resulting understanding helps analyzing the practical phenomena encountered when using actors and critics represented as neural networks. From this new light, we revisit in Section 5 a few existing algorithms whose components provide an alternative to the building blocks involved in the undesirable cyclic convergence process, and we suggest alternative solutions to these issues.

## 2 Related work

Issues when combining RL with function approximation have been studied for a long time (Baird & Klopf, 1993; Boyan & Moore, 1995; Tsitsiklis & Van Roy, 1997). In particular, it is well known that

deep RL algorithms can diverge when they meet three conditions coined as the "deadly triad" (Sutton & Barto, 2018), that is when they use (1) function approximation, (2) bootstrapping updates and (3) off-policy learning. However, these questions are mostly studied in the continuous state, discrete action case. For instance, several recent papers have studied the mechanism of this instability using DQN (Mnih et al., 2013). In this context, four failure modes have been identified from a theoretical point of view by considering the effect of a linear approximation of the deep-Q updates and by identifying conditions under which the approximate updates of the critic are contraction maps for some distance over Q-functions (Achiam et al., 2019). Meanwhile, van Hasselt et al. (2018) shows that, due to its stabilizing heuristics, DQN does not diverge much in practice when applied to the ATARI domain.

In contrast to these papers, here we study a failure mode specific to continuous action actor-critic algorithms. It hinges on the fact that one cannot take the maximum over actions, and must rely on the actor as a proxy for providing the optimal action instead. Therefore, the failure mode identified in this paper cannot be reduced to any of the ones that affect DQN. Besides, the theoretical derivations provided in the appendices show that the failure mode we are investigating does not depend on function approximation errors, thus it cannot be directly related to the deadly triad.

More related to our work, several papers have studied failure to gather rewarded experience from the environment due to poor exploration (Colas et al., 2018; Fortunato et al., 2017; Plappert et al., 2017), but we go beyond this issue by studying a case where the reward is actually found but not properly exploited. Finally, like us the authors of Fujimoto et al. (2018a) study a failure mode which is specific to DDPG-like algorithms, but the studied failure mode is different. They show under a batch learning regime that DDPG suffers from an *extrapolation error* phenomenon, whereas we are in the more standard incremental learning setting and focus on a deadlock resulting from the shape of the Q-function in the sparse reward case.

## 3 BACKGROUND: DEEP DETERMINISTIC POLICY GRADIENT

The DDPG algorithm (Lillicrap et al., 2015) is a deep RL algorithm based on the Deterministic Policy Gradient theorem (Silver et al., 2014). It borrows the use of a replay buffer and target networks from DQN (Mnih et al., 2015). DDPG is an instance of the Actor-Critic model. It learns both an actor function $\pi_\psi$ (also called policy) and a critic function $Q_\theta$, represented as neural networks whose parameters are respectively noted $\psi$ and $\theta$.

The deterministic actor takes a state $s \in S$ as input and outputs an action $a \in A$. The critic maps each state-action pair $(s, a)$ to a value in $\mathbb{R}$. The reward $r : S \times A \to \mathbb{R}$, the termination function $t : S \times A \to \{0, 1\}$ and the discount factor $\gamma < 1$ are also specified as part of the environment.

The actor and critic are updated using stochastic gradient descent on two losses $L_\psi$ and $L_\theta$. These losses are computed from mini-batches of samples $(s_i, a_i, r_i, t_i, s_{i+1})$, where each sample corresponds to a transition $s_i \to s_{i+1}$ resulting from performing action $a_i$ in state $s_i$, with subsequent reward $r_i = r(s_i, a_i)$ and termination index $t_i = t(s_i, a_i)$.

Two target networks $\pi_{\psi'}$ and $Q_{\theta'}$ are also used in DDPG. Their parameters $\psi'$ and $\theta'$ respectively track $\psi$ and $\theta$ using exponential smoothing. They are mostly useful to stabilize function approximation when learning the critic and actor networks. Since they do not play a significant role in the phenomena studied in this paper, we ignore them in the formal proofs given in appendices.

Equations (1) and (2) define $L_\psi$ and $L_\theta$:

$$L_\psi = - \sum_i Q_\theta \left( s_i, \pi_\psi \left( s_i \right) \right) \tag{1}$$

$$\begin{cases} \forall i, y_i = r_i + \gamma(1 - t_i)Q_{\theta'} \left( s_{i+1}, \pi_{\psi'} \left( s_{i+1} \right) \right) \\ L_\theta = \sum_i \left[ Q_\theta \left( s_i, a_i \right) - y_i \right]^2. \end{cases} \tag{2}$$

Training for the loss given in (1) yields the parameter update in (3), with $\alpha$ the learning rate:

$$\psi \leftarrow \psi + \alpha \sum_i \frac{\partial \pi_\psi(s_i)}{\partial \psi}^T \nabla_a Q_\theta(s_i, a)|_{a = \pi_\psi(s_i)}. \tag{3}$$

As DDPG uses a replay buffer, the mini-batch samples are acquired using a behaviour policy $\beta$ which may be different from the actor $\pi$. Usually, $\beta$ is defined as $\pi$ plus a noise distribution, which in the case of DDPG is either a Gaussian function or the more sophisticated Ornstein-Uhlenbeck noise.

Importantly for this paper, the behaviour of DDPG can be characterized as an intermediate between two extreme regimes:

- When the actor is updated much faster than the critic, the policy becomes greedy with respect to this critic, resulting into a behaviour closely resembling that of the Q-LEARNING algorithm. When it is close to this regime, DDPG can be characterized as off-policy.

- When the critic is updated much faster than the actor, the critic tends towards $Q^\pi(s, a)$. The problems studied in this paper directly come from this second regime.

A more detailed characterization of these two regimes in given in Appendix A.

## 4    A NEW FAILURE MODE

In this section, we introduce a simplistic environment which we call 1D-TOY. It is a one-dimensional, discrete-time, continuous state and action problem, depicted in Figure 1.

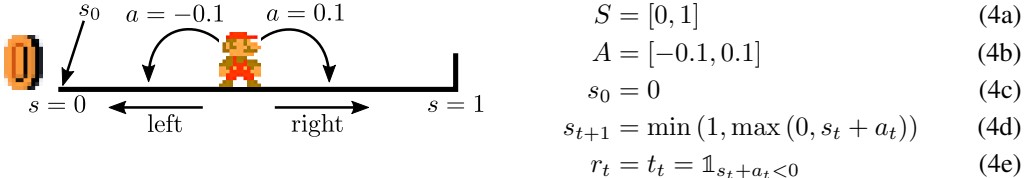

$$S = [0, 1] \tag{4a}$$
$$A = [-0.1, 0.1] \tag{4b}$$
$$s_0 = 0 \tag{4c}$$
$$s_{t+1} = \min\left(1, \max\left(0, s_t + a_t\right)\right) \tag{4d}$$
$$r_t = t_t = \mathbb{1}_{s_t + a_t < 0} \tag{4e}$$

Figure 1: The 1D-TOY environment

Despite its simplicity, DDPG can fail on 1D-TOY. We first show that DDPG fails to reach $100\%$ success. We then show that if learning a policy does not succeed soon enough, the learning process can get stuck. Besides, we show that the initial actor can be significantly modified in the initial stages before finding the first reward. We explain how the combination of these phenomena can result into a deadlock situation. We generalize this explanation to any deterministic and sparse reward environment by revealing and formally studying a undesirable cyclic process which arises in such cases. Finally, we explore the consequences of getting into this cyclic process.

### 4.1    EMPIRICAL STUDY

In all experiments, we set the maximum episode length $N$ to 50, but the observed phenomena persist with other values.

**Residual failure to converge using different noise processes**    We start by running DDPG on the 1D-TOY environment. This environment is trivial as one infinitesimal step to the left is enough to obtain the reward, end the episode and succeed, thus we might expect a quick $100\%$ success. However, the first attempt using an Ornstein-Uhlenbeck (OU) noise process shows that DDPG succeeds in only $94\%$ of cases, see Figure 2a.

These failures might come from an exploration problem. Indeed, at the start of each episode the OU noise process is reset to zero and gives little noise in the first steps of the episode. In order to remove this potential source of failure, we replace the OU noise process with an exploration strategy similar to $\epsilon$-greedy which we call "probabilistic noise". For some $0 < p < 1$, with probability $p$, the action is randomly sampled (and the actor is ignored), and with probability $1 - p$ no noise is used

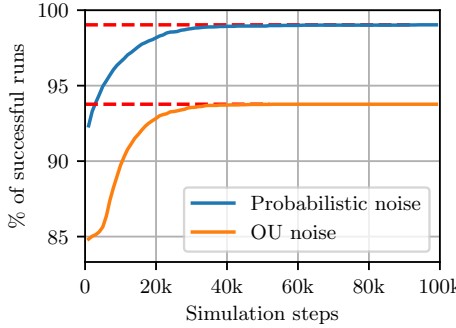 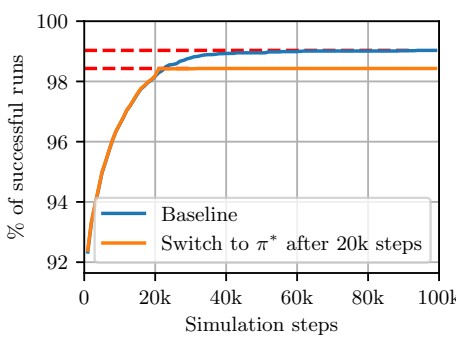

(a) Success rate of DDPG with Ornstein-Uhlenbeck (OU) and probabilistic noise. Even with probabilistic noise, DDPG fails on about 1% of the seeds.

(b) Comparison between DDPG with probabilistic noise and a variant in which the behaviour policy is set to the optimal policy $\pi^*$ after 20k steps.

Figure 2: Success rate of variants of DDPG on 1D-TOY over learning steps, averaged over 10k seeds. More details on learning algorithm and success evaluation are given in Appendix E.

and the raw action is returned. In our tests, we used $p = 0.1$. This guarantees at least a 5% chance of success at the first step of each episode, for any policy. Nevertheless, Figure 2a shows that even with probabilistic noise, about 1% of seeds still fail to converge to a successful policy in 1D-TOY, even after 100k training steps. All the following tests are performed using probabilistic noise.

We now focus on these failures. On all failing seeds, we observe that the actor has converged to a saturated policy that always goes to the right ($\forall s, \pi(s) = 0.1$). However, some mini-batch samples have non-zero rewards because the agent still occasionally moves to the left, due to the probabilistic noise applied during rollouts. The expected fraction of non-zero rewards is slightly more than 0.1%[1]. Figure 3a shows the occurrence of rewards in minibatches taken from the replay buffer when training DDPG on 1D-TOY. After each rollout (episode) of $n$ steps, the critic and actor networks are trained $n$ times on minibatches of size 100. So for instance, a failed episode of size 50 is followed by a training on a total of 5000 samples, out of which we expect more than 5 in average are rewarded transitions. More details about the implementation are available in Appendix E.

The constant presence of rewarded transitions in the minibatches suggests that the failures of DDPG on this environment are not due to insufficient exploration by the behaviour policy.

**Correlation between finding the reward early and finding the optimal policy** We have shown that DDPG can get stuck in 1D-TOY despite finding the reward regularly. Now we show that when DDPG finds the reward early in the training session, it is also more successful in converging to the optimal policy. On the other hand, when the first reward is found late, the learning process more often gets stuck with a sub-optimal policy.

From Figure 3b, the early steps appear to have a high influence on whether the training will be successful or not. For instance, if the reward is found in the first 50 steps by the actor noise (which happens in 63% of cases), then the success rate of DDPG is 100%. However, if the reward is first found after more than 50 steps, then the success rate drops to 96%. Figure 3b shows that finding the reward later results in lower success rates, down to 87% for runs in which the reward was not found in the first 1600 steps. Therefore, we claim that there exists a critical time frame for finding the reward in the very early stages of training.

**Spontaneous actor drift** At the beginning of each training session, the actor and critic of DDPG are initialized to represent respectively close-to-zero state-action values and close-to-zero actions. Besides, as long as the agent does not find a reward, it does not benefit from any utility gradient. Thus we might expect that the actor and critic remain constant until the first reward is found. Actually, we

---

[1]10% of steps are governed by probabilistic noise, of which at least 2% are the first episode step, of which 50% are steps going to the left and leading to the reward.

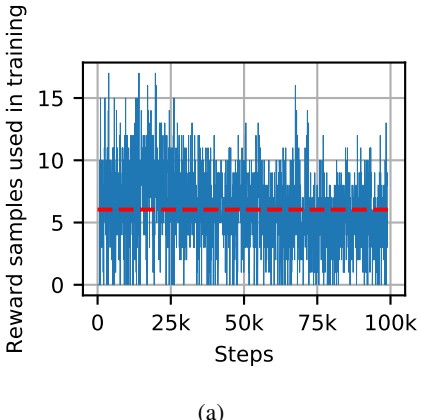
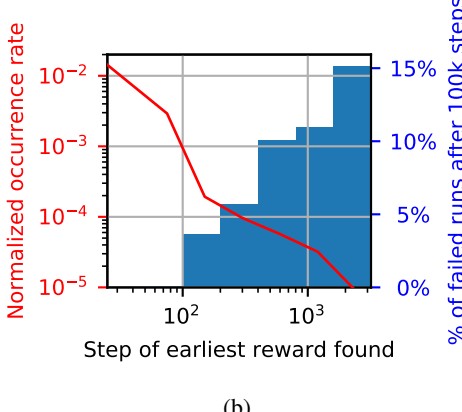

(a)                                                  (b)

Figure 3: (a) Number of rewards found in mini-batches during training. After a rollout of $n$ steps, the actor and critic are both trained on $n$ minibatches of size 100. The red dotted line indicates an average of 6.03 rewarded transitions present in these $n$ minibatches. (b) In red, normalized probability of finding the earliest reward at this step. In blue, for each earliest reward bin, fraction of these episodes that fail to converge to a good actor after 100k steps. Note that when the reward is found after one or two episodes, the convergence to a successful actor is certain.

show that even in the absence of reward, training the actor and critic triggers non-negligible updates that cause the actor to reach a saturated state very quickly.

To investigate this, we use a variant of 1D-TOY called DRIFT where the only difference is that no rewarded or terminal transitions are present in the environment. We also use a stripped-down version of DDPG, removing rollouts and using random sampling of states and actions as minibatches for training.

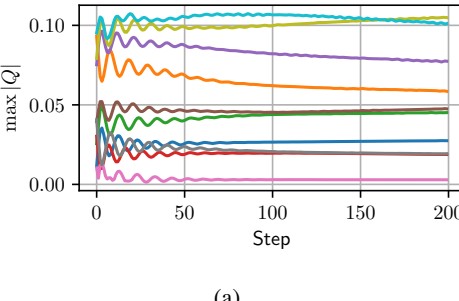
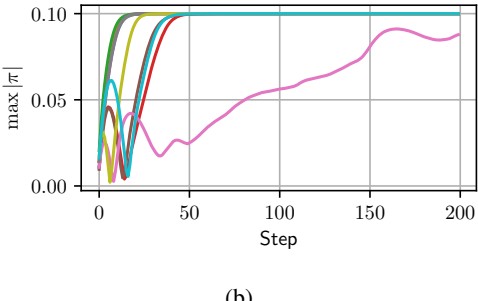

(a)                                                  (b)

Figure 4: Drift of $\max |Q|$ and $\max |\pi|$ in the DRIFT environment, for 10 different seeds. In the absence of reward, the critic oscillates briefly before stabilizing. However, the actor very quickly reaches a saturated state, at either $\forall s, \pi(s) = 0.1$ or $-0.1$.

Figure 4b shows that even in the absence of reward, the actor function drifts rapidly (notice the horizontal scale in steps) to a saturated policy, in a number of steps comparable to the "critical time frame" identified above. The critic also has a transitive phase before stabilizing.

In Figure 4a, the fact that $\max_{s,a} |Q(s,a)|$ can increase in the absence of reward can seem counter-intuitive, since in the loss function presented in Equation (2), $|y_i|$ can never be greater than $\max_{s,a} |Q(s,a)|$. However, it should be noted that the changes made to $Q$ are not local to the minibatch points, and increasing the value of $Q$ for one input $(s, a)$ may cause its value to increase for other inputs too, which may cause an increase in the global maximum of $Q$. This phenomenon is at the heart of the over-estimation bias when learning a critic (Fujimoto et al., 2018b), but this bias does not play a key role here.

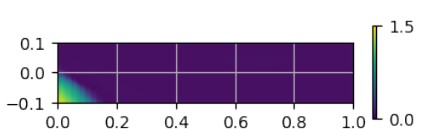
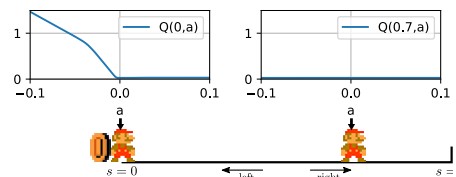
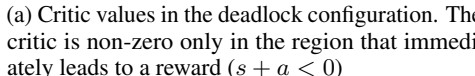

(a) Critic values in the deadlock configuration. The critic is non-zero only in the region that immediately leads to a reward ($s + a < 0$)

(b) Two snapshots of the critic for different states in a failed run. The high $Q$ values in the $s + a < 0$ region are not propagated.

Figure 5: Visualization of the critic in a failing run, in which the actor is stuck to $\forall s, \pi(s) = 0.1$.

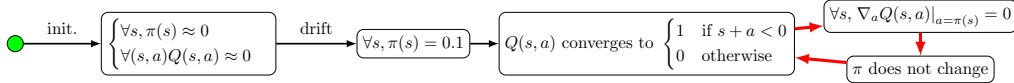

Figure 6: Deadlock observed in 1D-TOY, represented as the cycle of red arrows.

## 4.2 EXPLAINING THE DEADLOCK SITUATION FOR DDPG ON 1D-TOY

Up to now, we have shown that DDPG fails about $1\%$ of times on 1D-TOY, despite the simplicity of this environment. We have now collected the necessary elements to explain the mechanisms of this deadlock in 1D-TOY.

Figure 5 shows the value of the critic in a failed run of DDPG on 1D-TOY. We see that the value of the reward is not propagated correctly outside of the region in which the reward is found in a single step $\{(s, a) \mid s + a < 0\}$. The key of the deadlock is that once the actor has drifted to $\forall s, \pi(s) = 0.1$, it is updated according to $\nabla_a Q_\theta(s, a)|_{a=\pi_\psi(s)}$ (Equation (3)). Figure 5b shows that for $a = \pi(s) = 0.1$, this gradient is zero therefore the actor is not updated. Besides, the critic is updated using $y_i = r(s_i, a_i) + \gamma Q(s_i', \pi(s_i'))$ as a target. Since $Q(s_i', 0.1)$ is zero, the critic only needs to be non-zero for directly rewarded actions, and for all other samples the target value remains zero. In this state the critic loss given in Equation (2) is minimal, so there is no further update of the critic and no further propagation of the state-action values. The combination of the above two facts clearly results in a deadlock.

Importantly, the constitutive elements of this deadlock do not depend on the batches used to perform the update, and therefore do not depend on the experience selection method. We tested this experimentally by substituting the behaviour policy for the optimal policy after 20k training steps. Results are presented in Figure 2b and show that, once stuck, even when it is given ideal samples, DDPG stays stuck in the deadlock configuration. This also explains why finding the reward early results in better performance. When the reward is found early enough, $\pi(s_0)$ has not drifted too far, and the gradient of $Q(s_0, a)$ at $a = \pi(s_0)$ drives the actor back into the correct direction.

Note however that even when the actor drifts to the right, DDPG does not always fail. Indeed, because of function approximators the shape of the critic when finding the reward for the first time varies, and sometimes converges slowly enough for the actor to be updated before the convergence of the critic.

Figure 6 summarizes the above process. The entry point is represented using a green dot. First, the actor drifts to $\forall s, \pi(s) = 0.1$, then the critic converges to $Q^\pi$ which is a piecewise-constant function (Experiment in Figure 5, proof in Theorem 1 in Appendix B), which in turn means that the critic provides no gradient, therefore the actor is not updated (as seen in Equation 3, more details in Theorem 2) [2].

## 4.3 GENERALIZATION

Our study of 1D-TOY revealed how DDPG can get stuck in this simplistic environment. We now generalize to the broader context of more general continuous action actor critic algorithms, including

---

[2] Note that Figure 5 shows a critic state which is slightly different from the one presented in Figure 6, due to the limitations of function approximators.

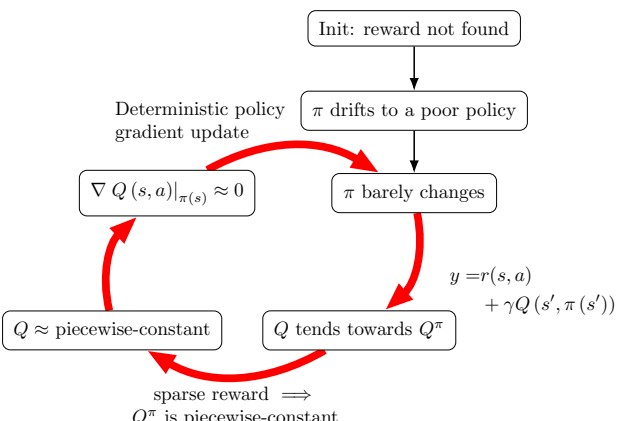

Figure 7: A cyclic view of the undesirable convergence process in continuous action actor-critic algorithms, in the deterministic and sparse reward case.

at least DDPG and TD3, and acting in any deterministic and sparse reward environment. The generalized deadlock mechanism is illustrated in Figure 7 and explained hereafter in the idealized context of perfect approximators, with formal proofs rejected in appendices.

**Entry point:**   As shown in the previous section, before the behaviour policy finds any reward, training the actor and critic can still trigger non-negligible updates that may cause the actor to quickly reach a poor state and stabilize. This defines our entry point in the process.

**Q tends towards $Q^\pi$:**   A first step into the cycle is that, if the critic is updated faster than the policy, the update rule of the critic $Q$ given in Equation (2) makes $Q$ converge to $Q^\pi$. This is presented in detail in Appendix C.

**$Q^\pi$ is piecewise-constant:**   In Appendix D, we then show that, in a deterministic environment with sparse terminal rewards, $Q^\pi$ is piecewise-constant because $V^\pi(s')$ only depends on two things: the (integer) number of steps required to reach a rewarded state from $s'$, and the value of this reward state, which is itself piecewise-constant. Note that we can reach the same conclusion with non-terminal rewards, by making the stronger hypothesis on the actor that $\forall s, r(s, \pi(s)) = 0$. Notably, this is the case for the actor $\forall s, \pi(s) = 0.1$ on 1D-TOY.

**Q is approximately piecewise-constant and $\nabla_{\mathbf{a}} \mathbf{Q}(\mathbf{s}, \mathbf{a})|_{\mathbf{a}=\pi(\mathbf{s})} \approx \mathbf{0}$:**   Quite obviously, from $Q^\pi$ is piecewise-constant and $Q$ tends towards $Q^\pi$, we can infer that $Q$ progressively becomes almost piecewise-constant as the cyclic process unfolds. Actually, the $Q$ function is estimated by a function approximator which is never truly discontinuous. The impact of this fact is studied in Section 4.5. However, we can expect $Q$ to have mostly flat gradients since it is trained to match a piecewise-constant function. We can thus infer that, globally, $\nabla_a Q(s, a)|_{a=\pi(s)} \approx 0$. And critically, the gradients in the flat regions far from the discontinuities give little information as to how to reach regions of higher values.

**$\pi$ barely changes:**   DDPG uses the deterministic policy gradient update, as seen in Equation (3). This is an analytical gradient that does not incorporate any stochasticity, because $Q$ is always differentiated exactly at $(s, \pi(s))$. Thus the actor update is stalled, even when the reward is regularly found by the behaviour policy. This closes the loop of our process.

### 4.4   CONSEQUENCES OF THE CONVERGENCE CYCLE

As illustrated with the red arrows in Figure 7, the more loops performed in the convergence process, the more the critic tends to be piecewise-constant and the less the actor tends to change. Importantly, this cyclic convergence process is triggered as soon as the changes on the policy drastically slow down or stop. What matters for the final performance is the quality of the policy reached before

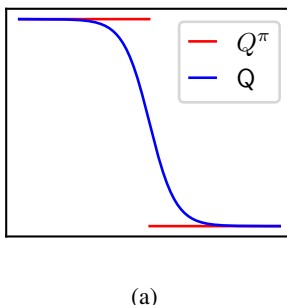
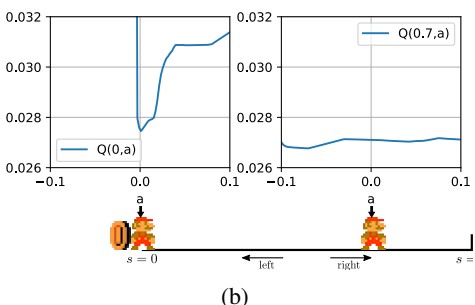

(a)                                                              (b)

Figure 8: (a) Example of a monotonous function approximator. (b) Simply changing the vertical scale of the graphs presented in Figure 5b reveals that the function approximator is not perfectly flat, and has many unwanted local extrema. Specifically, continuously moving from $\pi(0) = 0.1$ to $\pi(0) < 0$ requires crossing a significant valley in $Q(0, a)$, while $\pi(0) = 0.1$ is a strong local maximum.

this convergence loop is triggered. Quite obviously, if the loop is triggered before the policy gets consistently rewarded, the final performance is deemed to be poor.

The key of this undesirable convergence cycle lies in the use of the deterministic policy gradient update given in Equation (3). Actually, rewarded samples found by the exploratory behaviour policy $\beta$ tend to be ignored by the conjunction of two reasons. First, the critic is updated using $Q(s', \pi(s'))$ and not $Q(s, \beta(s))$, thus if $\pi$ differs too much from $\beta$, the values brought by $\beta$ are not properly propagated. Second, the actor being updated through (3), i.e. using the analytical gradient of the critic with respect to the actions of $\pi$, there is no room for considering other actions than that of $\pi$. Besides, the actor update involves only the state $s$ of the sample taken from the replay buffer, and not the reward found from this sample $r(s, a)$ or the action performed. For each sample state $s$, the actor update is intended to make $\pi(s)$ converge to $\mathrm{argmax}_a \pi(s, a)$ but the experience of different actions performed for identical or similar states is only available through $Q(s, \cdot)$, and in DDPG it is only exploited through the gradient of $Q(s, \cdot)$ at $\pi(s)$, so the process can easily get stuck in a local optimum, especially if the critic tends towards a piecewise-constant function, which as we have shown happens when the reward is sparse. Besides, since TD3 also updates the actor according to (3) and the critic according to (2), it is susceptible to the same failures as DDPG.

### 4.5 IMPACT OF FUNCTION APPROXIMATION

We have just explained that when the actor has drifted to an incorrect policy before finding the reward, an undesirable convergence process should result in DDPG getting stuck to this policy. However, in 1D-TOY, we measured that the actor drifts to a policy moving to the right in $50\%$ of cases, but the learning process only fails $1\%$ of times. More generally, despite the issues discussed in this paper, DDPG has been shown to be efficient in many problems. This better-than-predicted success can be attributed to the impact of function approximation.

Figure 8a shows a case in which the critic approximates $Q^\pi$ while keeping a monotonous slope between the current policy value and the reward. In this case, the actor is correctly updated towards the reward (if it is close enough to the discontinuity). This is the most often observed case, and naturally we expect approximators to smooth out discontinuities in target functions in a monotonous way, which facilitates gradient ascent. However, the critic is updated not only in state-action pairs where $Q^\pi(s, a)$ is positive, but also at points where $Q^\pi(s, a) = 0$, which means that the bottom part of the curve also tends to flatten. As this happens, we can imagine phenomena that are common when trying to approximate discontinuous functions, such as the overshoot observed in Figure 8b. In this case, the gradient prevents the actor from improving.

## 5 POTENTIAL SOLUTIONS

In the previous section, we have shown that actor-critic algorithms such as DDPG and TD3 could not recover from early convergence to a poor policy due to the combination of three factors whose

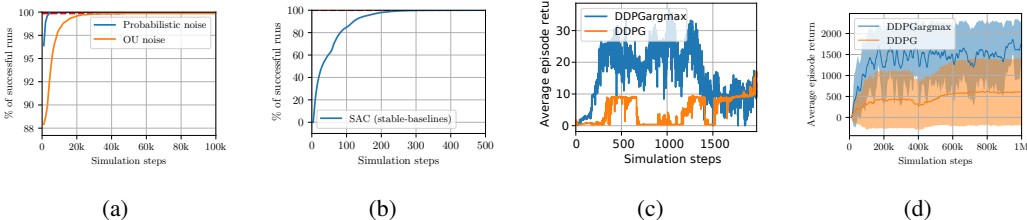

Figure 9: (a) Applying DDPG-argmax to 1D-TOY. (b) Applying SAC to 1D-TOY. In both cases, the success rate reaches 100% quickly. (c) Applying DDPG and DDPG-argmax to a sparse-reward variant of the REACHER-V2 environment. (d) Applying DDPG and DDPG-argmax to a sparse-reward variant of the HALFCHEETAH-V2 environment. Details on the changes made to REACHER-V2 and HALFCHEETAH-V2 are available in Appendix F.2.

dependence is highlighted in Figure 7: the use of the deterministic policy gradient update, the use of $Q(s', \pi(s'))$ in the critic update, and the attempt to address sparse reward in deterministic environments. In this section, we categorize existing or potential solutions to the above issue in terms of which of the above factor they remove.

**Avoiding sparse rewards:**    Transforming a sparse reward problem into a dense one can solve the above issue as the critic should not converge to a piecewise-constant function anymore. This can be achieved for instance by using various forms of shaping (Konidaris & Barto, 2006) or by adding auxiliary tasks (Jaderberg et al., 2016; Riedmiller et al., 2018). We do not further investigate these solutions here, as they are mainly problem-dependent and may introduce bias when the reward transformation results in deceptive gradient or modifies the corresponding optimal policy.

**Replacing the policy-based critic update:**    As explained above, if some transition $(s, a, s')$ leading to a reward is found in the replay buffer, the critic update corresponding to this transition uses $Q(s', \pi(s'))$, therefore not propagating the next state value that the behaviour policy may have found. Of course, when using the gradient from the critic, the actor update should tend to update $\pi$ to reflect the better policy such that $\pi(s') \to a'$, but the critic does not always provide an adequate gradient as shown before.

If performing a maximum over a continuous action space was possible, using $\max_a Q(s', a)$ instead of $Q(s', \pi(s'))$ would solve the issue. Several works start from this insight. Some methods directly sample the action space and look for such an approximate maximum (Kalashnikov et al., 2018; Simmons-Edler et al., 2019). To show that this approach can fix the above issue, we applied it to the 1D-TOY environment. We take a straightforward implementation where the policy gradient update in DDPG is replaced by sampling 100 different actions, finding the argmax over these actions of $Q(s, a)$, and regressing the actor towards the best action we found. We call the resulting algorithm DDPG-argmax, and more details are available in Appendix F.1. Results are shown in Figure 9a, in which we see that the success rate quickly reaches 100%.

Quite obviously, even if sampling can provide a good enough baseline for simple enough benchmarks, these methods do not scale well to large actions spaces. Many improvements to this can be imagined by changing the way the action space is sampled, such as including $\pi(s)$ in the samples, to prevent picking a worse action than the one provided by the actor, sampling preferentially around $\pi(s)$, or around $\pi(s + \epsilon)$, or just using actions taken from the replay buffer.

Interestingly, using a stochastic actor such as in the Soft Actor Critic (SAC) algorithm (Haarnoja et al., 2018a;b) can be considered as sampling preferentially around $\pi(s + \epsilon)$ where $\epsilon$ is driven by the entropy regularization term. In Figure 9b, we show that SAC also immediately solves 1D-TOY.

Another approach relies on representing the critic as the $V$ function rather than the $Q$ function. The same way $\pi(s)$ tends to approximate $\text{argmax}_a Q(s, a)$, $V$ tends to approximate $\max_a Q(s, a)$, and is updated when finding a transition that raises the value of a state. Using $V$, performing a maximum in the critic update is not necessary anymore. The prototypical actor-critic algorithm using a model of $V$ as a critic is CACLA (Van Hasselt & Wiering, 2007). However, approximating $V$ with neural

networks can prove more unstable than approximating $Q$, as function approximation can be sensitive to the discontinuities resulting form the implicit maximization over $Q$ values.

**Replacing the deterministic policy gradient update:**   Instead of relying on the deterministic policy gradient update, one can rely on a stochastic policy to perform a different actor update. This is the case of SAC, as mentioned just above. Because SAC does not use $Q(s', \pi(s'))$ in its update rule, it does not suffer from the undesirable convergence process described here.

Another solution consists in completely replacing the actor update mechanism, using regression to update $\pi(s)$ towards any action better than the current one. This could be achieved by updating the actor and the critic simultaneously: when sampling a higher-than-expected critic value $y_i > Q(s_i, a_i)$, one may update $\pi(s_i)$ towards $a_i$ using:

$$L_\psi = \sum_i \delta_{y_i > Q(s_i, \pi(s_i))} \left( \pi(s_i) - a_i \right). \tag{5}$$

This is similar to the behaviour of CACLA, as analyzed in Zimmer & Weng (2019).

**Larger benchmarks**   Whether the deadlock situation investigated so far occurs more in more complex environments is an important question. To investigate this, we performed additional experiments based on more complex environments, namely sparse versions of REACHER-V2 and HALFCHEETAH-V2. Results are depicted in Figure 9c and 9d and more details are presented in Appendix F.2. One can see that DDPG-argmax outperforms DDPG, which seems to indicate that the failure mode we are studying is also at play. However, with higher-dimensional and more complex environments, the analysis becomes more difficult and other failures modes such as the ones related to the deadly triad, the extrapolation error or the over-estimation bias might come into play, so it becomes harder to quantitatively analyze the impact of the phenomenon we are focusing on. On one hand, this point showcases the importance of using very elementary benchmarks in order to study the different failure modes in isolation. On the other hand, trying to sort out and quantify the impact of the different failure modes in more complex environments is our main objective for future work.

## 6    CONCLUSION AND FUTURE WORK

In RL, continuous action and sparse reward environments are challenging. In these environments, the fact that a good policy cannot be learned if exploration is not efficient enough to find the reward is well-known and trivial. In this paper, we have established the less trivial fact that, if exploration does find the reward consistently but not early enough, an actor-critic algorithm can get stuck into a configuration from which rewarded samples are just ignored. We have formally characterized the reasons for this situation and we have outlined existing and potential solutions. Beyond this, we believe our work sheds new light on the convergence regime of actor-critic algorithms.

Our study was mainly built on a simplistic benchmark which made it possible to study the revealed deadlock situation in isolation from other potential failure modes such as exploration issues, the over-estimation bias, extrapolation error or the deadly triad. The impact of this deadlock situation in more complex environments is a pressing question. For this, we need to sort out and quantify the impact of these different failure modes. Using new tools such as the ones provided in Ahmed et al. (2019), recent analyses of the deadly triad such as Achiam et al. (2019) as well as simple, easily visualized benchmarks and our own tools, for future work we aim to conduct deeper and more exhaustive analysis of all the instability factors of DDPG-like algorithms, with the hope to contribute in fixing them.

## 7    ACKNOWLEDGEMENTS

Anonymized for submission.

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

# A    Two regimes of DDPG

In this section, we characterize the behavior of DDPG as an intermediate between two extremes, that we respectively call the *critic-centric view*, where the actor is updated faster, resulting in an algorithm close to Q-LEARNING, and the *actor-centric view*, where the critic is updated faster, resulting in a behaviour more similar to Policy Gradient.

## A.1    Actor update: the critic-centric view

The Q-LEARNING algorithm (Watkins, 1989) and its continuous state counterpart DQN (Mnih et al., 2013) rely on the computation of a policy which is greedy with respect to the current critic at every time step, as they simply take the maximum of the Q-values over a set of discrete actions. In continuous action settings, this amounts to an intractable optimization problem if the action space is large and nontrivial.

We get a simplified vision of DDPG by considering an extreme regime where the actor updates are both fast enough and good enough so that $\forall s, \pi(s) \approx \mathrm{argmax}_a Q(s, a)$. We call this the *critic-centric* vision of DDPG, since the actor updates are assumed to be ideal and the only remaining training is performed on the critic.

In this regime, by replacing $\pi(s)$ with $\mathrm{argmax}_a Q(s, a)$ in Equation (2), we get $y_i = r_i + \gamma(1 - t_i) \max_a Q(s_i, a)$, which corresponds to the update of the critic in Q-LEARNING and DQN. A key property of this regime is that, since the update in based on a maximum over actions, the resulting algorithm is truly off-policy. We can thus infer that most of the off-policiness property of DDPG comes from keeping it close to this regime.

## A.2    Critic update: the actor-centric view

Symmetrically to the previous case, if the critic is updated well and faster than the actor, it tends to represent the critic of the current policy, $Q^\pi$. Furthermore, if the actor tends to change slowly enough, critic updates can be both fast and good enough so that it reaches the fixed point of the Bellman equation, that is $\forall (s, a, r, t, s'), Q(s, a) = r + \gamma(1 - t)Q(s', \pi(s'))$.

In this case, the optimization performed in (1) mostly consists in updating the actor so that it exploits the corresponding critic by applying the deterministic policy gradient on the actor. This gives rise to a *actor-centric* vision of DDPG.

# B    Deadlock in 1D-toy

In this section, we prove that there exists a state of DDPG that is a deadlock in the 1D-TOY environment. This proof directly references Figure 6. Let us define two functions $Q$ and $\pi_\psi$ such that:

$$\forall (s, a), Q(s, a) = \mathbb{1}_{s+a<0} \tag{6}$$
$$\forall s \in S, \pi_\psi(s) = 0.1 \tag{7}$$

From now on, we will use notation $\pi := \pi_\psi$.

**Theorem 1.** $(Q, \pi_\psi)$ is a fixed point for the critic update.

*Proof.* The critic update is governed by Equation 2.

Let $(s_i, a_i, r_i, t_i, s_i')$ be a sample from the replay buffer. The environment dictates that $r_i = t_i = \mathbb{1}_{s_i+a_i<0}$.

$$y_i = r_i + \gamma(1 - t_i)Q\left(s_i', \pi_\psi(s_i')\right)$$
$$y_i = r_i + \gamma(1 - t_i)Q\left(s_i', 0.1\right) \qquad \text{by (7)}$$
$$y_i = r_i \qquad \text{by (6)}$$
$$y_i = \mathbb{1}_{s_i+a_i<0}$$
$$y_i = Q(s_i, a_i) \qquad \text{by (6)}$$

Therefore, for each sample $y_i = Q(s_i, a_i)$, and $L$ is null and minimal. Therefore $\theta$ will not be updated during the critic update. $\square$

**Theorem 2.** $(Q, \pi_\psi)$ is a fixed point for the actor update.

*Proof.* The actor update is governed by Equation 3.

Let $\{(s_i, a_i, r_i, t_i, s_i')\}$ be a set of samples from the replay buffer. The environment dictates that $\forall i, r_i = t_i = \mathbb{1}_{s_i + a_i < 0}$.

$$\psi \leftarrow \psi + \alpha \sum_i \frac{\partial \pi_\psi(s_i)}{\partial \psi}^T \nabla_a Q(s_i, a)|_{a = \pi_\psi(s_i)}$$

Since $Q(s_i, a) = \mathbb{1}_{s_i + a < 0}$, $\nabla_a Q(s_i, a) = 0$, so $\psi$ will not be updated during the actor update. $\square$

In this section, we assume that $Q$ is any function, however in implementations $Q$ is often a parametric neural network $Q_\theta$, which cannot be discontinuous. Effects of this approximation are discussed in Section 4.5.

## C    CONVERGENCE OF THE CRITIC TO $Q^\pi$

**Notation**    For a state-action pair $s, a$, we define $s_1$ as the result of applying action $a$ at state $s$ in the deterministic environment. For a given policy $\pi$, we define $a_1$ as $\pi(s_1)$. Recursively, for any $i \geq 1$, we define $s_{i+1}$ as the result of applying action $a_i$ to state $s_i$, and $a_i$ as $\pi(s_i)$.

**Definition 1.** Let $(s, a) \in S \times A$. If $(s, a)$ is terminal, then we set $N = 0$. Otherwise, we set $N$ to the number of subsequent transitions with policy $\pi$ to reach a terminal state. Therefore, the transition $(s_N, a_N)$ is always terminal. We generalize by setting $N = \infty$ when no terminal transition is ever reached.

We define the state-action value function of policy $\pi$ as:

$$Q^\pi(s, a) := r(s, a) + \sum_{i=1}^N \gamma^i r(s_i, a_i)$$

Note that when $N = \infty$, the sum converges under the hypothesis that rewards are bounded and $\gamma < 1$.

If $\pi$ is fixed, $Q$ is updated regularly via approximate dynamic programming with the Bellman operator for the policy $\pi$. Under strong assumptions, or assuming exact dynamic programming, it is possible to prove (Geist & Pietquin (2011)) that the iterated application of this operator converges towards a unique function $Q^\pi$, which corresponds to the state-action value function of $\pi$ as defined above. It is usually done by proving that the Bellman operator is a contraction mapping, and also applies in deterministic cases.

However, when using approximators such as neural nets, no theroretical results of convergence exists, to the best of our knowledge. In this paper we assume that this convergence is true, and in the experimental results we did not observe any failures to converge towards $Q^\pi$. On the contrary, we observe that this converge occurs, and can be what starts the deadlock cycle studied in Section 4.3.

## D    PROOF THAT $Q^\pi$ IS PIECEWISE-CONSTANT

In this section, we show that in deterministic environments with terminal sparse rewards, $Q^\pi$ is piecewise-constant.

**Definition 2.** In this article, for $I \subset \mathbb{R}^n$, we say a function $f : I \to \mathbb{R}$ is piecewise-constant if $\forall x_0 \in I$, either $\nabla_x f(u)|_{x=x_0} = 0$, or $f$ has no gradient at $x_0$.

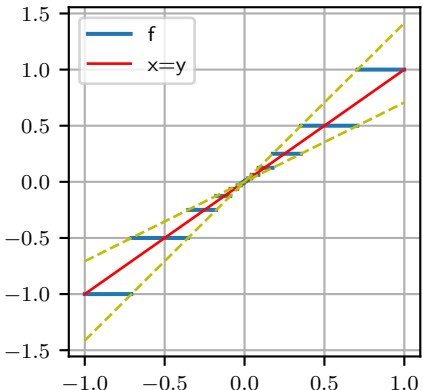

Figure 10: Example of a non-continuous function $f$ with values in $\left\{\left(\frac{1}{2}\right)^n \mid n \in \mathbb{N}\right\}$, approximating the identity function. However, this function is not differentiable because the difference quotient does not converge but instead oscillates between two values.

**Theorem 3.** In a deterministic environment with terminal sparse rewards, for any $\pi$, $Q^\pi$ is piecewise-constant.

*Proof.* Note that this proof can be trivialized by assuming that around any point where the gradient is defined, there exists a neighbourhood in which the function is continuous. In this case, the intermediate value theorem yields an uncountable set of values of the function in this neighbourhood, which contradicts the countable number of possible discounted rewards.

The crux of the following proof is that even when no such neighbourhood exists, the gradient is either null of non-existent. This behavior is shown in Figure 10.

Using the notations of Definition 1 and the theorem hypothesis that rewarded transitions are also terminal, we can write $Q^\pi(s, a)$ as:

$$Q^\pi(s, a) = \begin{cases} r(s, a) & \text{if } N = 0 \\ \gamma^N r(s_N, a_N) & \text{if } N \text{ is finite} \\ 0 & \text{otherwise.} \end{cases}$$

We promote $N$ to a function $S \times A \to \mathbb{N} \cup \{+\infty\}$, and we define a function $u : S \times A \to \mathbb{R}$ as given in Equation (8).

$$u(s, a) = \begin{cases} r(s, a) & \text{if } N = 0 \\ r\left(s_{N(s,a)}, a_{N(s,a)}\right) & \text{if } N > 0 \text{ finite} \\ 0 & \text{otherwise.} \end{cases} \tag{8}$$

Now we have $\forall (s, a) \in S \times A, Q^\pi(s, a) = \gamma^{N(s,a)} u(s, a)$.

Let $R$ be the finite set of possible reward values.

Therefore values of $Q^\pi$ are in a set $M = \{\gamma^n r \mid n \in \mathbb{N}, r \in R\}$. Let $M^+ = M \cap \mathbb{R}^{+*}$ be the set of positive values of $M$. Since $R \subset \mathbb{R}$ is finite, we order all non-zero positive possible rewards in increasing order $r_1, r_2, \cdots r_k$.

Let $M_k^+ = \{\gamma^n r \mid n \in \mathbb{N}, r \in R_k\}$ where $R_k = \{r_1, \cdots, r_k\}$.

We prove the following by recurrence over the number of possible non-zero rewards:

$$H(k) : \exists \nu_k > 0, \forall \delta > 0, \exists \text{ consecutive } b, a \in M_k^+, \delta \nu_k < a - b \text{ and } b < a < \delta$$

**Initialization**   When $k = 1$, $M^+ = \{r_1\gamma^n \mid n \in \mathbb{N}\}$. Let $\nu = \frac{\gamma^2}{1-\gamma}$. Let $\delta > 0$. Let $n = \lfloor \log_\gamma \frac{\delta}{r_1} \rfloor + 1$. We have:

$$\log_\gamma \frac{\delta}{r_1} - 1 < n - 1 \leq \log_\gamma \frac{\delta}{r_1}$$

$$\log_\gamma \frac{\delta}{r_1} < n < \log_\gamma \frac{\delta}{r_1} + \log_\gamma(1-\gamma) + 2 - \log_\gamma(1-\gamma)$$

$$\log_\gamma \frac{\delta}{r_1} < n < \log_\gamma \frac{\delta(1-\gamma)}{r_1} + \log_\gamma \nu$$

$$\log_\gamma \frac{\delta}{r_1} < n < \log_\gamma \frac{\delta\nu(1-\gamma)}{r_1}$$

$$\frac{\delta\nu(1-\gamma)}{r_1} < \gamma^n < \frac{\delta}{r_1}$$

$$\delta\nu(1-\gamma)^2 < r_1\gamma^n(1-\gamma) < \delta(1-\gamma)$$

$$\delta\nu < r_1\gamma^n - r_1\gamma^{n+1} \text{ and } r_1\gamma^n < \delta$$

Let $a = r_1\gamma^n \in M^+$ and $b = r_1\gamma^{n+1} \in M^+$. $\delta\nu < a - b$ and $b < a < \delta$ therefore $H(1)$ is verified.

**Recurrence**   Let $k \geq 1$, and assume $H(k)$ is true. Let $\nu_k$ be the $\nu$ chosen for $H(k)$. Let $\nu_{k+1} = \frac{\nu_k}{2}$. Let $\delta > 0$. Let $b_k, a_k$ a consecutive pair chosen in $M_k^+$ such that $\delta\nu_k < a_k - b_k$ and $b_k < a_k < \delta$.

Since $R_{k+1}$ contains only one more element than $R_k$, which is larger than all elements in $R_k$, we know that there is either one or zero elements $c \in M_{k+1}^+$ that are strictly between $a_k$ and $b_k$. If $a_k - c < c - b_k$ then let $a_{k+1} = c$ and $b_{k+1} = b_k$, otherwise $a_{k+1} = a_k$ and $b_{k+1} = c$. If $a_k$ and $b_k$ are still consecutive in $M_{k+1}^+$, then $a_{k+1} = a_k$ and $b_{k+1} = b_k$.

This guarantees that $[b_{k+1}, a_{k+1}]$ as at least half as big as $[b_k, a_k]$. Therefore, $\frac{1}{2}(a_k - b_k) < a_{k+1} - b_{k+1}$, which means that $\delta\nu_{k+1} < a_{k+1} - b_{k+1}$ and $b_k < a_k < \delta$.

Therefore $H(k+1)$ is verified.

This also gives the general expression of $\nu$, valid for all $k$: $\nu = \left(\frac{\gamma^2}{1-\gamma}\right)^{|R|}$.

**Main proof**   Using the result above, we prove that $Q^\pi(s, a)$ cannot have any non-null derivatives.

Trivially, $Q^\pi$ cannot have a non-null derivative at a point $(s, a)$ where $Q^\pi(s, a) = q_0 \neq 0$. Indeed, there exists a neighbourhood of $q_0 \in M$ in which there is a single value.

Let $x_0 = (s, a)$ such that $Q(s, a) = 0$. Let $v$ be a vector of the space $S \times A$. Let $f : \mathbb{R} \to \mathbb{R}$ be defined as $f(h) = Q^\pi(x_0 + hv)$. In the following, we show that $\frac{f(h)}{|h|}$ cannot converge to a non-null value when $h \to 0$.

We use the $(\epsilon, \delta)$ definition of limit. If $f$ had a non-null derivative $l$ at 0, we owould have $\forall \epsilon > 0, \exists \delta > 0, \forall h, |h| < \delta \implies \left|\frac{f(h)}{|h|} - l\right| < \epsilon$.

Instead, we will show the opposite: $\exists \epsilon > 0, \forall \delta > 0, \exists h, |h| < \delta$ and $\left|\frac{f(h)}{|h|} - l\right| \geq \epsilon$.

Using the candidate derivative $l$ and the $\nu$ value computed above that only depends on $\gamma$ and $|R|$, we set $\epsilon = \frac{l\nu}{2}$.

Let $\delta > 0$.

There exists consecutive $b, a$ in $M$ such that $\delta l\nu \leq a - b$ and $b < a < \delta l$.

We set $h = \frac{a+b}{2l}$. Note that $\frac{a+b}{2} < \delta l$ therefore $h < \delta$.

$f(h)$ is in $M$, but $hl$ is the center of the segment $[b, a]$ of consecutive points of $M$. Therefore, the distance between $f(h)$ and $hl$ is at least $\frac{a-b}{2}$.

$$|f(h) - hl| \geq \frac{a-b}{2} \geq \frac{\delta l \nu}{2}$$

Since $h < \delta$, $\frac{1}{h} > \delta$.

$$\left| \frac{f(h)}{h} - l \right| \geq \frac{l\nu}{2} = \epsilon.$$

□

# E  IMPLEMENTATION DETAILS

Here is the complete rollout and training algorithm, taken from the Spinup implementation of DDPG.

**Result:** Policy $\pi_\psi$, number of steps before success
$\pi_\psi, Q_\theta \leftarrow$ Xavier uniform initializer
env_steps $\leftarrow 0$
**for** $t \leftarrow 1$ **to** $10000$ **do**
    $a \leftarrow \pi_\psi(s)$
    **if** $rand() < 0.1$ **then**
        $a \leftarrow \text{rand}(-0.1, 0.1)$
    **end**
    Step the environment using action $a$, get a transition $(s, a, r, t, s')$
    Store $(s, a, r, t, s')$ in the replay buffer
    env_steps $\leftarrow$ env_steps $+ 1$
    **if** $t = 1$ *or env_steps* $> N$ **then**
        Reset the environment
        **for** $k \leftarrow 1$ **to** *env_steps* **do**
            Sample a mini-batch of size 100 from the replay buffer
            Train $\pi_\psi$ and $Q_\theta$ on this replay buffer with losses (1) and (2).
        **end**
        env_steps $\leftarrow 0$
    **end**
    **if** $t \bmod 1000 = 0$ **then**
        **if** *last 20 episodes were successes* **then**
            Terminate the algorithm, and return success_after_steps$= t$.
        **end**
    **end**
**end**

# F  PROPOSED SOLUTION TO THE DEADLOCK PROBLEM

## F.1  DESCRIPTION OF DDPG-ARGMAX

In this paper, we identified a deadlock problem and tracked its origin to the actor update described in Equation (1). In Section 5, we proposed a new actor update mechanism in an algorithm called DDPG-argmax, which we describe in more details here.

Instead of relying on the differentiation of $Q_\theta(s_i, \pi_\psi(s_i))$ to update $\psi$ in order to maximize $Q(s, \pi(s))$, we begin by selecting a set of $N$ potential actions $(b_j)_{0 \leq j < N}$. Then, we compute $Q_\theta(s_i, b_j)$ for each sample $s_i$ and each potential action $b_j$, and for each sample $s_i$ we find the best potential action $c_i = b_{\mathrm{argmax}_j Q_\theta(s_i, b_j)}$. Finally, we regress $\pi_\psi(s_i)$ towards the goal $c_i$. This process is summarized in Equation (9), where $Unif(A)$ stands for uniform sampling in $A$.

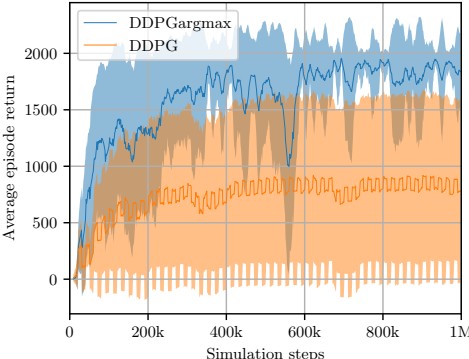

Figure 11: Performance of DDPG and DDPG-argmax on a sparse variant of HALFCHEETAH-V2. To ensure exploration of the state space is not a problem, the policy is replaced with a good pre-trained policy in one episode over 20.

$$\begin{cases} (b_j)_{0 \leq j < N} \sim \mathrm{Unif}(A) \\ \quad\quad c_i = b_{\mathrm{argmax}_j\, Q_\theta(s_i, b_j)} \\ \mathrm{minimize} \sum_i \left( \pi_\psi(s_i) - c_i \right)^2 \text{ w.r.t. } \quad \psi \end{cases} \tag{9}$$

### F.2 EXPERIMENTS ON LARGER BENCHMARKS

In order to test the relevance of using DDPG-argmax on larger benchmarks, we constructed sparse reward versions of REACHER-V2 and HALFCHEETAH-V2.

REACHER-V2 was modified by generating a step reward of 1 when the distance between the arm and the target is less than $0.06$, and 0 otherwise. The distance to the target was removed from the observations, and the target was fixed to a position of $[0.19, 0]$, instead of changing at each episode. We also removed the control penalty.

HALFCHEETAH-V2 was modified by generating a step reward of 2 when the x component of the speed of the cheetah is more than 2. We also removed the control penalty. Since the maximum episode duration is 1000, the maximum possible reward in this modified environment is 2000.

In both cases, the actor noise uses the default implementation of the Spinup implementation of DDPG, which is an added uniform noise with an amplitude of $0.1$.

Running DDPG and DDPG-argmax on these environments yields the results shown in Figures 9c and 9d. Experiments on HALFCHEETAH-V2 have been conducted using six different seeds. In Figure 9d, the main curves are smoothed using a moving average covering 10 episodes (10k steps), and the shaded area represents the average plus or minus one standard deviation.

On HALFCHEETAH-V2, both DDPG and DDPG-argmax are able to find rewards despite its sparsity. However, DDPG-argmax outperforms DDPG in this environment. Since the only difference between these algorithms is the actor update, we conclude that even in complex environments, the actor update is the main weakness of DDPG. We have shown that replacing it with a brute-force update improves performance dramatically, and further research aiming to improve the performance of deterministic actor-critic algorithms in environments with sparse rewards should concentrate on improving the actor update rule.

Figure 9d shows that DDPG is able to find the reward without the help of any exploration except the uniform noise built in the algorithm itself. However, to prove that state-space exploration is not the issue here, we constructed a variant in which the current actor is backed up and replaced with a pre-trained good actor every 20 episodes. This variant achieves episode returns above 1950 (as a

reminder, the maximum episode return is 2000). In the next episode, the backed up policy is restored. This guarantees that the replay buffer always contains all the transitions necessary to learn a good policy. We call this technique *priming*.

Results of this variant are presented in Figure 11. Notice that DDPG performs much better than without priming, but the performance of DDPG-argmax is unchanged. However, DDPG still fails to completely solve the environment, proving that even when state-space exploration is made trivial, DDPG underperforms on sparse-reward environments due to its poor actor update.

