# OpenReview forum: "The problem with DDPG: understanding failures in deterministic environments with sparse rewards"
_ICLR.cc/2020/Conference — Reject_

### Official Review · AnonReviewer3 · 2019-10-20
**Official Blind Review #3**

**Rating:** 3

**Review:**

Summary:
This work studies the instability problem of DDPG in the setting of a deterministic
environment and sparse reward. This work designed a toy environment to showcase
the potential issues of leading to the instability of DDPG. The observations, such as the correlation between the early access of good trajectory leads to more stable performance later, the deadlock of training could be beneficial.
It is essential to analyze and understand the intrinsic properties of the classic algorithms, and it would benefit the research community a lot if the empirical study is appropriately designed and conducted. Overall, this paper studied an essential problem
of the vulnerability of the classic DRL algorithm (DDPG), which should attract more attention and efforts
from the research community.


Detailed comments:
Methodology:
The experiments conducted cannot support the conclusions in this paper.
I can not fully understand the conclusion from the subsection "Residual failure to converge using different noise processes". The DDPG agent is finding the reward regularly while it couldn't converge to the 100% optimal performance. In my opinion, this is an observation,
instead of giving any useful conclusion. The convergent issues when using the combination of off-policy
learning, function approximation, and bootstrapping are known (Sutton and Barto, Chap 11, 2018).

The increasing of Q value seems natural to me due to the overestimation of Q learning,
even with the zero reward setting, which is one of the motivations of Double DQN.

Several potential solutions are discussed while no empirical evidence or theoretical
justification is provided, even in the designed 1D-Toy example.

It would be more convincing that the conclusions can be validated on more challenging
tasks such as regular continuous action benchmarks (mujoco, etc.)

Writing:
The presentation of this work is not ready for publication, given its current form.
What is the definition of reward function? The formula as shown in Eq 4e is not clear.
What is the optimal performance of 1D-TOY example?

**Experience Assessment:**

I have read many papers in this area.

**Review Assessment: Checking Correctness Of Derivations And Theory:**

I did not assess the derivations or theory.

**Review Assessment: Checking Correctness Of Experiments:**

I assessed the sensibility of the experiments.

**Review Assessment: Thoroughness In Paper Reading:**

I read the paper at least twice and used my best judgement in assessing the paper.

---

> ### Author Response · Authors · 2019-11-11
> **Response to Reviewer #3**
>
> We thank the reviewer for his/her nice comments about the importance of analyzing and understanding the intrinsic properties of classic algorithms.
>
> However, our feeling is that the reviewer has missed an important point of our paper. Indeed, the reviewer reminds that the convergence issues when using the combination of off-policy learning, function approximation, and bootstrapping are known (Sutton and Barto, Chap 11, 2018). We agree, this is known as the deadly triad, and we spend the first paragraph of the related work section describing some recent works about this issue, then explaining that the phenomenon we are studying is different and specific to the continuous action case (while the deadly triad also occurs in the discrete action case). An additional element to convince the reviewer that the phenomenon we are studying is not the deadly triad is the following. Function approximation is one of the three components of the deadly triad. Without function approximation, the "deadly triad convergence issue" disappears. In our case, the theoretical derivation of the deadlock phenomenon does not involve function approximation, on the contrary function approximation tends to mitigate the fundamental issue. We added this further element into the related work section (see p. 2).
>
> The reviewer mentions that he/she can not fully understand the conclusion from the subsection "Residual failure to converge using different noise processes". The point of this subsection is just to establish that even in the simplistic 1d-toy environment, there are cases where DDPG fails and cannot recover from these failures. We believe the local study supports this very local conclusion. The broader conclusion of the paper is on the mechanisms explaining this failure, and is backed up by the next sections and the proofs we provide in the appendices.
>
> About the definition of the reward function, it is defined in Fig. 1 (Eq. 4e), as the rest of the MDP. In a sentence, it means that the agent gets a reward of 1 if its next action moves it beyond the left extremity of the platform, 0 otherwise.
>
> The optimal performance in 1D-toy is obtained by always moving left from the start (s=0) position. The agent always gets a reward of 1 in one step.
>
> About the increasing of Q-values, the reviewer is right. The fact that Q-values can increase even if the absence of reward is due to a combination of non-zero initialization, function approximation and the over-estimation bias. Actually the comment the reviewer is referring to just serves to explain that Q-values can change as it appears in Fig.4a, but as Fig.4a shows, these values do not tend to increase much under the zero reward regime. Thus the over-estimation bias does not play a detrimental role here. We clarified what the paragraph was referring to and reminded the link to the over-estimation bias. The whole paragraph could be turned into a footnote or removed if the reviewers think it is not necessary.
>
> About the fact that no empirical evidence or theoretical justification is provided to the solutions we listed on 1D-toy, we agree. This is now fixed, see the answers to the other reviewers. Same comment about validating the conclusions on more challenging environments: we agree, we added new experiments using more complex environments, as also requested by the other reviewers.

---

### Official Review · AnonReviewer2 · 2019-10-22
**Official Blind Review #2**

**Rating:** 6

**Review:**

Overview: This paper describes a shortfall with the DDPG algorithm on a continuous state action space with sparse rewards. To first prove the existence of this shortfall, the authors demonstrate its theoretical possibility by reviewing the behavior of DDPG actor critic equations and the “two-regimes” proofs in the appendices. They then demonstrate the occurrence of the critic being updated faster than the actor, leading to a sub-optimal convergence from which the model can never recover. In this demonstration, they use a very simple environment they created, “1D-Toy”. The 1D-Toy environment is a one-dimensional, discrete-time, continuous state and action problem. Moving to the left at all in 1D-Toy results in a reward and episode end. Episode length was set at 50, as the agent could move to the right forever and never stop the episode. The authors demonstrate how the failure of the agent to obtain 100% success in this simple environment was, in fact, due to the phenomenon mentioned earlier. If the agent managed to obtain a reward very early on in training, it was highly likely the agent would converge on an optimal solution. If not, the actor would drift to a state were it no longer updates, and the critic would similarly no longer update either, resulting in a deadlock and suboptimal policy. The authors then generalize their findings using a helpful figure (Figure 7) which describes the cyclical nature of the phenomenon and how it can happen in any environment. Finally, the authors mention potential solutions to prevent the training failure from occurring, such as avoiding sparse rewards, replacing the critic update to avoid loss, etc.

Contributions: the discovery and review of a potential training failure for DDPG due to the nature of the critic update being reliant on the policy, and the deterministic policy gradient update

Questions and Comments:
I believe that this work is relevant to ICLR and to the field. The paper is well-written, the theory is sound, and the experiment is sufficient to describe the stated deadlock situation that DDPG can contain during training. I believe this paper should be accepted because of these reasons. I have a few comments/questions for the authors which I have written below.

I’m interested to see how likely this deadlock situation is on more complex environments. Did you run experiments on common benchmarks and analyze them?
You mention several potential solutions, one of them being the avoidance of sparse rewards. Of course this is problem-dependent which you stated yourself. The other two involve replacing two of the update functions. In the policy-based critic update, both of the mentioned solutions in this section have drawbacks mentioned. Would using a stochastic policy gradient update affect the networks ability to learn successfully in more complex environments? Would this make training less stable?

I’m curious to see in which directions you see this work being extended. You briefly mention in the conclusion that there would be more formal studies: how do you imagine these being?


**Experience Assessment:**

I have read many papers in this area.

**Review Assessment: Checking Correctness Of Derivations And Theory:**

I assessed the sensibility of the derivations and theory.

**Review Assessment: Checking Correctness Of Experiments:**

I assessed the sensibility of the experiments.

**Review Assessment: Thoroughness In Paper Reading:**

I read the paper at least twice and used my best judgement in assessing the paper.

---

> ### Author Response · Authors · 2019-11-11
> **Response to Reviewer #2**
>
> We warmly thanks the reviewer for his/her positive appreciation of our paper and for providing useful directions on how to improve our work.
>
> The reviewer is right that whether the deadlock situation we outlined occurs more in more complex environments is an important question. To investigate this, we first designed a new algorithm called DDPG-argmax where instead of applying the PG, we sample N actions, get the argmax over these actions of Q(s, a) and regress the policy towards the best action we found. DDPG-argmax should be immune to the deadlock failure mode that we are outlining in the paper. Then we performed additional experiments based on more complex environments, namely sparsified versions of Reacher-v2 and HalfCheetah-v2. The results (Fig. 9c and 9d) show that DDPG-argmax clearly outperforms DDPG, though both algorithms consistently find some reward, which means that the gain in performance does not come exclusively from improved exploration. Thus we consider that these results support the assumption that the deadlock failure mode is also at play in these larger benchmarks.
>
> However, we have to note that, with higher-dimensional and more complex environments, the analysis becomes more difficult and other failures modes such as the ones related to the deadly triad or the maximization bias can come into play, so it becomes harder to quantitatively analyze the impact of the phenomenon we are focusing on. Trying to sort out and quantify the impact of the different failure modes in more complex environments is our main objective for future work.
>
> The reviewer asks whether a stochastic policy gradient update is immune to our deadlock phenomenon. First, the notion of "stochastic policy gradient update" is ambiguous. Does the reviewer mean a gradient update based on a stochastic policy, or a stochastic gardient update applied to a deterministic (or even stochastic) policy? As the second interpretation raises some issues, we opted for the first and tried SAC on our 1D-toy environment. Our results in Fig. 9b show that SAC reaches 100% success, thus it is indeed immune to the deadlock effect in this simplistic environment.
>
> So it seems that using a stochastic policy can fix the problem. However, we would like to also find a solution in the case of deterministic policies. The main difference between the last version of SAC and TD3 is that the latter uses a deterministic policy, and several recent empirical studies have shown that these two algorithms outperform eachother depending on the environment (see e.g. "Ahmed et al. Understanding the Impact of Entropy on Policy Optimization (ICML 2019)" for the beginning of an analysis of this fact). So we believe that trying to successfully learn deterministic policies remains a mandatory endeavour.
>
> About our future work, as we mentioned above, we believe we should try to sort out and quantify the impact of different failure modes in more complex environments. Using new tools such as the ones provided in Ahmed et al. 2019, recent analyses of the deadly triad such as Achiam et al. 2019 as well as simple, easily visualized benchmarks and our own tools, we aim to conduct deeper and more exhaustive analysis of all the instability factors of DDPG-like algorithms, with the hope to contribute in fixing them. We have added a future work paragraph in the conclusion to better account for this perspective (see p. 10).

---

### Official Review · AnonReviewer1 · 2019-10-22
**Official Blind Review #1**

**Rating:** 3

**Review:**

The paper investigates why DDPG can sometimes fail in environments with sparse rewards. It presents a simple environment that helps the reader build intuition and supports the paper's empirical investigation. First, the paper shows that DDPG fails on the simple environment in ~6% of cases, despite the solution being trivial—go left from the start state. The paper then augments DDPG with epsilon-greedy-style exploration to see if the cause of these failures is simply inadequate exploration. Surprisingly, in 1% of cases DDPG still fails. The paper shows that even in these failure cases there were still rewarded transitions that could have been learned from, and investigates relationships between properties of individual runs and the likelihood of failure. The paper then explains how these failures occur: the policy drifts to always going right, and the critic converges to a piecewise constant function whose gradient goes to zero and prevents further updates to the policy. The paper then generalizes this deadlock mechanism to other continuous-action actor-critic algorithms like TD3 and discusses how function approximation helps mitigate this issue. Finally, the paper gives a brief overview of some existing potential solutions.

Currently, I recommend rejecting this paper; while it is very well-written and rigorously investigates a problem with a popular algorithm, the paper does not actually present a novel solution method. It does a great job of clearly defining and investigating a problem and shows how others have attempted to solve it in the past, but stops there. It doesn't propose a new method and/or compare the existing methods empirically, nor does it recommend a specific solution method.

A much smaller concern is that the problem investigated is somewhat niche; it happens in a very small percentage of runs and is mitigated by the type of function approximation commonly used. This lowers its potential impact a little.

The introduction does a great job of motivating the paper. I would've liked the related work section to elaborate more on the relationships between the insights in this paper and those of Fujimoto et al. (2018a), since the paper says the insights are related. Section 3 gave a very clear overview of DDPG. The simple environment described in section 4 was intuitive and explained well, and the empirical studies were convincing.

However, after section 4 established the existence of the deadlock problem in DDPG, I was expecting the rest of the paper to present a solution method and empirically validate it. Instead, section 5 generalizes the findings from section 4 to include other environments and algorithms. I felt that section 4 and 5 could have been combined and shortened, and the extra space used to explain and empirically test a novel solution method. For example, figures 6 and 7 seem to be conveying similar information, but figure 7 takes up almost half a page.

Currently this paper seems like a good candidate for a workshop. It convincingly establishes the existence of a problem and shows what causes the problem to occur, but doesn't contribute a solution to the problem. Submitting it to a workshop could be a good opportunity for discussion, feedback, and possible collaboration, all of which might help inspire a solution method.

**Experience Assessment:**

I have read many papers in this area.

**Review Assessment: Checking Correctness Of Derivations And Theory:**

I did not assess the derivations or theory.

**Review Assessment: Checking Correctness Of Experiments:**

I assessed the sensibility of the experiments.

**Review Assessment: Thoroughness In Paper Reading:**

I read the paper at least twice and used my best judgement in assessing the paper.

---

> ### Author Response · Authors · 2019-11-11
> **Response to Reviewer #1**
>
> We thank the reviewer for the very accurate summary of our paper. Our work was completely understood and the points put forward by the reviewer definitely helped us strengthen our work.
>
> Nevertheless, the reviewer "currently recommends rejecting the paper ... [based on the fact that] it does not actually present a novel solution method" (this is repeated at the end of the same paragraph and of the review). We would like to draw his/her attention on the fact that there are many well cited papers whose main contribution is to analyze the mechanisms of a problem, and all of them do not present a novel solution. For instance, based on the above criterion, one would reject John N. Tsitsiklis and Benjamin Van Roy's "Analysis of Temporal-Difference Learning with Function Approximation", NIPS 1997 (1260 citations).
>
> Furthermore, we could argue that we actually suggested novel solution methods, with the following paragraph, after mentionning sampling the action space and taking the max (p. 9): "Many improvements to this can be imagined by changing the way the action space is sampled, such as including \pi(s) in the samples, to prevent picking a worse action than the one provided by the actor, sampling preferentially around \pi(s), or around \pi(s + \epsilon), or just using actions taken from the replay buffer."
>
> Anyways, we agree that the paper could be stronger with more focus on potential solutions. Thus, in the new Section 5, we strengthened the analysis and empirical comparison of existing and novel solutions to the problem we outlined. In particular, we checked that Soft Actor Critic (SAC) could easily solve the 1D-toy environment, and we investigated a novel solution called DDPG-argmax where instead of applying the PG, we sample N actions, get the argmax over these actions of Q(s, a) and regress the policy towards the best action we found. We show that this method solves the 1D-toy environment, and that it outperforms DDPG on sparse versions the 2-dimensional Reacher environment and the larger benchmark Half-Cheetah (see Fig. 9c and 9d). Nevertheless, we are aware that this proposed solution may still be unsatisfactory, e.g. in terms of scalability as larger benchmarks will require more expansive sampling.
>
> About the fact that the problem we outlined might be niche, we assume that the low probability of occurrence increases with the complexity of the agent-environment interaction, and that the deadlock mechanism we outlined might be a key factor of the widely recognized instability of DDPG-like algorithms in sparse reward environments. In the hope to backup this assumption, we performed the additional experiments based on the sparse versions of the larger environments mentioned above. The fact that DDPG-argmax outperforms DDPG although both can find some reward is a good sign that the deadlock mechanism could be at play. But we have to note that, the analysis being more difficult, it is harder to make sure that no other failure modes are involved here, as we stress in our conclusion. This point also strongly advocates for using very elementary benchmarks as we did to analyze isolated failure modes.
>
> About the relationship between our insights and those of Fujimoto et al. (2018a), it reduces to the fact that we both study a failure mode which is specific to DDPG-like algorithms. They show under a batch learning regime that DDPG suffers from an extrapolation error phenomenon, whereas we are in the more standard incremental learning setting and focus on a deadlock resulting from the shape of the Q-function in the sparse reward case. This has been rewritten at the end of the related work section.
>
> Making the changes resulting from all the reviewers comments required extra space. We followed the reviewer's advice and combined and shortened Section 4, 5 and 6 to obtain the necessary space.

---

### Author Response · Authors · 2019-11-11
**General response to all reviewers**

We thank all the reviewers for their comments which helped us a lot to improve the paper. We have posted a first revised version where all the changes we made are highligthed in blue. Figures 9d and 11 are still preliminary and will be finished for the next revision, but the tendency is already clear. We hope that all reviewers will find the time to consider the new version and we are ready to perform further changes as long as time permits. For explanation of these changes, please see the individual responses below.

---

### Decision · Program_Chairs · 2019-12-19

**Decision:**

Reject

**Comment:**

This paper provides an extensive investigation of the robustness of Deep Deterministic Policy Gradient algorithm.

Papers providing extensive and qualitative empirical studies, illustrative benchmark domains, identification of problems with existing methods, and new insights can be immensely valuable, and this paper is certainly in this direction, if not quite there yet.

The vast majority of this paper investigates one deep learning algorithm in designed domain. There is some theory but it's relegated to the appendix. There are a few issues with this approach: (1) there is no concrete evidence that this is a general issue beyond the provided example (more on that below). (2) Even in the designed domain the problem is extremely rare. (3) The study and perhaps even the issue is only shown for one particular architecture (with a whole host of unspecified meta-parameter details). Why not just use SAC it works? DDPG has other issues, why is it of interest to study and fix this particular architecture? The motivation that it is the first and most popular algorithm is not well developed enough to be convincing. (4) There is really no reasoning to suggest that the particular 1D is representative or interesting in general.

The authors including Mujoco results to address #1. But the error bars overlap, its completely unclear if the baseline was tuned at all---this is very problematic as the domains were variants created by the authors. If DDPG was not tuned for the variant then the plots are not representative. In general, there are basically no implementation details (how parameters were tested, how experiments were conducted)or general methodological details given in the paper. Given the evidence provided in this paper its difficult to claim this is a general and important issue.

I encourage the authors to look at John Langfords hard exploration tasks, and broaden their view of this work general learning mechanisms.